# Early-life stress induces persistent astrocyte dysfunction associated with fear generalisation

**Mathias Guayasamin[1,2†], Lewis R Depaauw-Holt[1,2†], Ifeoluwa I Adedipe[1,2†], Ossama Ghenissa[1,2], Juliette Vaugeois[1,2], Manon Duquenne[1,2], Benjamin Rogers[1,2], Jade Latraverse-Arquilla[2], Sarah Peyrard[2], Anthony Bosson[2], Ciaran Murphy-Royal[1,2]***

[1]Département de Neurosciences, Université de Montréal, Montréal, Canada; [2]Centre de Recherche du Centre Hospitalier de l'Université de Montréal, Montréal, Canada

## eLife Assessment

This **important** article explores the impact of early-life stress (ELS) on adult brain and behaviour. The significance of the **convincing** findings is that they implicate regulation of non-neuronal cells in the development of brain and behavioural dysfunction associated with ELS. With an elegant combination of behavioural models, morphological and functional assessments using immunostaining, electrophysiology, and viral-mediated loss-of-function approaches, the authors report that astrocyte dysfunction plays a role in ELS responses. The work is of interest to a broad behavioural and cellular neuroscience audience.

**Abstract** Early-life stress can have lifelong consequences, enhancing stress susceptibility and resulting in behavioural and cognitive deficits. While the effects of early-life stress on neuronal function have been well-described, we still know very little about the contribution of non-neuronal brain cells. Investigating the complex interactions between distinct brain cell types is critical to fully understand how cellular changes manifest as behavioural deficits following early-life stress. Here, using male and female mice we report that early-life stress induces anxiety-like behaviour and fear generalisation in an amygdala-dependent learning and memory task. These behavioural changes were associated with impaired synaptic plasticity, increased neural excitability, and astrocyte hypofunction. Genetic perturbation of amygdala astrocyte function by either reducing astrocyte calcium activity or reducing astrocyte network function was sufficient to replicate cellular, synaptic, and fear memory generalisation associated with early-life stress. Our data reveal a role of astrocytes in tuning emotionally salient memory and provide mechanistic links between early-life stress, astrocyte hypofunction, and behavioural deficits.

*For correspondence:
ciaran.murphy-royal@umontreal.ca

†These authors contributed equally to this work

**Competing interest:** The authors declare that no competing interests exist.

## Introduction

Both human and rodent studies have provided compelling evidence for the long-lasting effects of adverse early-life experiences, highlighting enhanced susceptibility to subsequent stressors later in life (*Walker et al., 2017*; *Joëls and Baram, 2009*; *Lopez and Bagot, 2021*; *O'Donnell and Meaney, 2020*; *Meaney, 2001*). Early-life stress (ELS) has widespread effects across the brain and can influence many neural circuits including those involved in threat detection, emotion, cognitive processing, and reward-seeking behaviours (*Peña et al., 2017*; *Peña et al., 2019*; *Lesuis et al., 2018*; *Lesuis et al., 2019*). Research investigating the neurobiology of stress has predominantly focused upon

the effects on neurons and resulting behaviour (*Ramkumar et al., 2024*; *Lupien et al., 2009*; *Bains et al., 2015*; *Senst et al., 2016*; *Wamsteeker Cusulin et al., 2013*; *Füzesi et al., 2016*; *Daviu et al., 2020*; *Guadagno et al., 2018b*; *Guadagno et al., 2021*; *Guadagno et al., 2018a*). We have gleaned tremendous insight into disorders of stress from these studies, yet the links between stress, brain circuits, and behaviour remain tenuous. Within this framework, the contribution of distinct brain cells, including astrocytes, remains largely untested. Glial cells, which comprise approximately half the cell population in the brain, have been shown to directly regulate synaptic transmission and plasticity (*Murphy-Royal et al., 2015*; *Murphy-Royal et al., 2020*; *Lau et al., 2021*; *Henneberger et al., 2010*; *Papouin et al., 2017*; *Covelo and Araque, 2018*; *Panatier et al., 2011*; *Panatier et al., 2006*; *Matos et al., 2018*; *Lefton et al., 2024*; *Guttenplan et al., 2024*), thereby influencing many discrete behaviours (*Nagai et al., 2021*; *Wahis et al., 2021*; *Robin et al., 2018*; *Martin-Fernandez et al., 2017*; *Yu et al., 2018*; *Nagai et al., 2019*; *Doron et al., 2022*; *Adamsky et al., 2018*; *Kol et al., 2020*; *Mu et al., 2019*; *Chen et al., 2024*). Despite research suggesting important roles for astrocytes in regulating affective states (*Wahis et al., 2021*; *Martin-Fernandez et al., 2017*), fear (*Martin-Fernandez et al., 2017*), reward and addiction behaviours (*Doron et al., 2022*; *Kruyer et al., 2022*; *Kruyer et al., 2019*), whether astrocyte dysfunction prompts stress-induced behavioural impairments remains poorly understood.

First, we characterised whether ELS influenced stress hormone levels across our experimental paradigm from pre-stress until young adulthood. We report that that ELS induces a transient increase in blood glucocorticoids that peaks during late adolescence/young adulthood. Turning to behaviour, this increase in glucocorticoid levels was associated with anxiety-like behaviour in the elevated plus maze.

Next, we set out to determine whether ELS influenced cognitive function using an amygdala-dependent threat discrimination assay. This task was specifically chosen to test the effects of ELS on amygdala function as this brain region is involved with threat detection and associative learning that is acutely sensitive to stress (*Guadagno et al., 2018b*; *Guadagno et al., 2021*; *Ressler, 2010*; *Malter Cohen et al., 2013*; *Roozendaal et al., 2009*). We employed an auditory discriminative fear conditioning paradigm that requires synaptic plasticity changes in the lateral amygdala (*Kim and Cho, 2017*) and found that ELS impairs discriminative fear memory, resulting in fear generalisation. These hormonal and behavioural changes were associated with impaired synaptic plasticity and enhanced excitability of cells in the lateral amygdala, evidenced by increased c-Fos labelling.

Finally, we investigated the impact of ELS on astrocytes in the lateral amygdala revealing changes in specific proteins associated with morphological reorganisation and astrocyte network function. ELS resulted in decreased frequency and amplitude in astrocyte calcium activity. To directly implicate astrocytes with behavioural dysfunction, we used two distinct viral approaches to mimic stress-induced astrocyte dysfunction specifically in the lateral amygdala. Impairment of astrocyte network function, or decreasing calcium activity, recapitulated ELS-induced fear generalisation, synaptic, and excitability phenotypes, supporting the hypothesis that astrocytes play an influential role in the effects of stress on neural circuits and behaviour. Together, these findings identify astrocytes as central elements regulating amygdala-dependent affective memory and highlight astrocytes as putative targets for the treatment of stress disorders.

## ELS increases blood corticosterone and induces anxiety-like behaviours

We employed an ELS paradigm which combines limited bedding and nesting materials as well as maternal separation from postnatal days 10–17 (*Figure 1A*) and has been shown to induce lifelong stress susceptibility (*Peña et al., 2017*). We first set out to determine the impact of this ELS paradigm on stress hormone levels across our experimental window. Taking trunk blood from mice across several litters at different time points in the ELS paradigm – pre-stress (P10), end of ELS paradigm (P17), at the start (P45), and end of our experimental window (P70) – we report that this ELS paradigm induces a latent increase in blood corticosterone with a significant increase occurring only in late adolescence P45, long after the termination of the stress paradigm (*Figure 1B*).

Next, we investigated whether this ELS paradigm results in long-term behavioural change, beginning with anxiety-like behaviours. We found that while there were no changes in behaviour in the open-field task (*Figure 1D and E*), we did see an increase in locomotor activity, that is, distance travelled, following ELS (*Figure 1F*). Using the elevated plus maze we, report increased anxiety-like behaviour with an increase in time spent in the closed arms and a reduction in time spent in the open

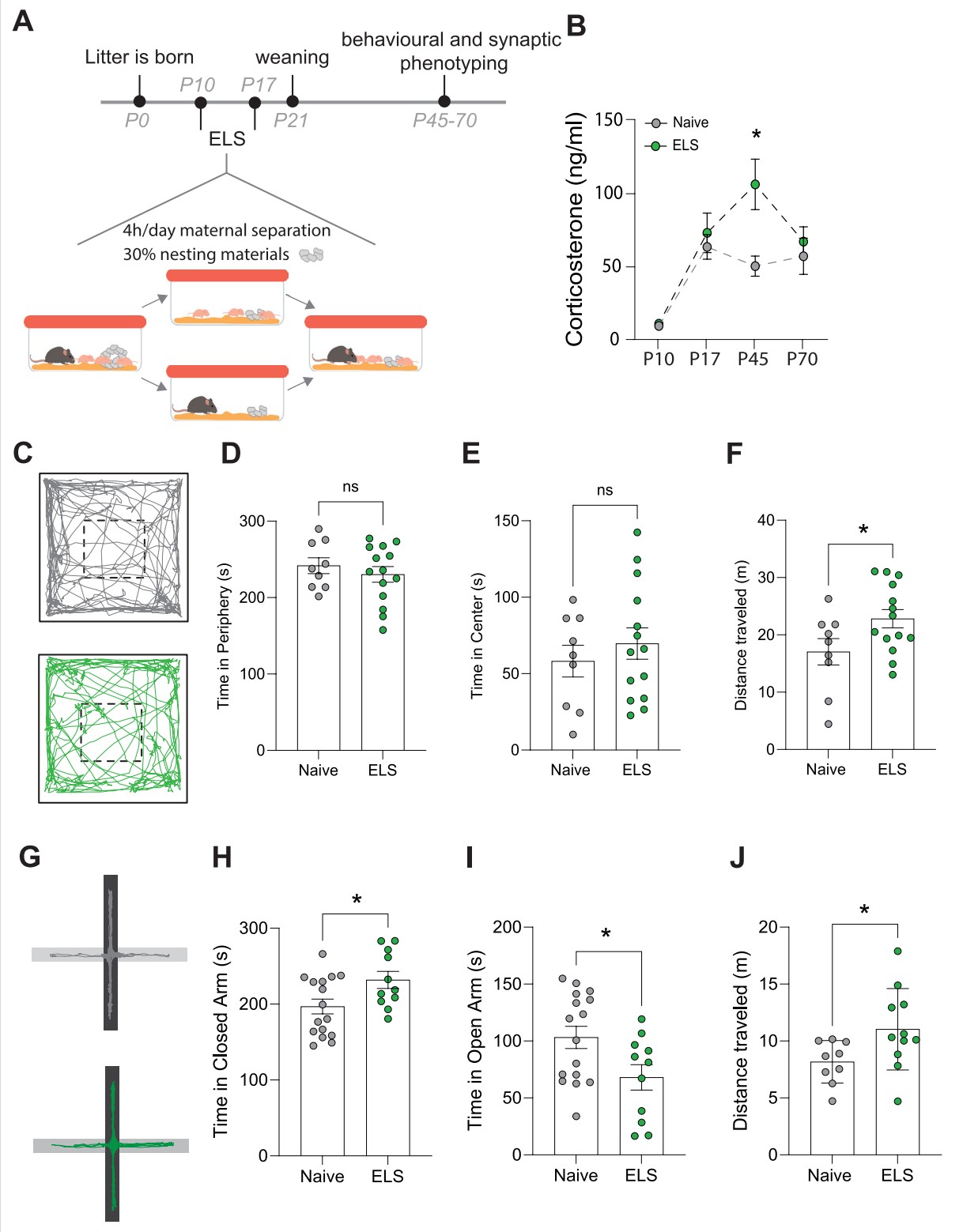

**Figure 1.** Effects of early-life stress (ELS) on anxiety-like behaviour. (**A**) Timeline of ELS and behavioural assays. (**B**) Serum corticosterone levels are increased during adolescence, following ELS. p=0.01, naïve N = 9, ELS N = 11 mice. (**C**) Representative movement tracks of naive (grey) and ELS (green) mice in open-field task. (**D**) No significant difference in time spent in the periphery of open-field maze following ELS. p=0.4591, naïve N = 9, ELS N = 14 mice. (**E**) No significant difference in time spent in the centre of open-field maze following ELS. p=0.4591, naïve N = 9, ELS N = 14 mice. (**F**) Total distance travelled during open-field test was increased in ELS mice. p=0.04, naïve N = 9, ELS N = 14 mice. (**G**) Representative locomotor tracks in

*Figure 1 continued on next page*

Figure 1 continued

elevated plus maze in naïve and ELS mice, black segment represents closed arms with grey arms representing open arms. (**H**) ELS mice had increased time spent in closed arms p=0.0276, naïve N = 18, ELS N = 11 mice. (**I**) ELS mice had decreased time spent in open arms p=0.0276, naïve N = 18, ELS N = 11 mice. (**J**) Total distance travelled during elevated plus maze test was increased by ELS. p=0.02, naïve N = 9, ELS N = 11 mice. See **Supplementary file 1** for detailed statistical summaries.

arms of the maze (**Figure 1H and I**). This was also accompanied by an increase in distance travelled within the maze (**Figure 1J**), similar to our findings in the open field. Increased locomotion in response to stress is consistent with previous reports (**Füzesi et al., 2016**; **Sharma et al., 2022**).

## Amygdala-dependent memory, plasticity, and excitability are disrupted by ELS

Considering the strong association between stress disorders, anxiety, and amygdala dysfunction, we next set out to determine whether amygdala-dependent behaviour was impacted by ELS. We specifically targeted lateral amygdala-dependent learning and memory using an auditory discriminative fear conditioning paradigm. In brief, during this behavioural task mice were first exposed to a neutral auditory cue (conditioned stimulus-; CS-) followed by a distinct auditory cue (CS+) paired with footshock (unconditioned stimulus; US). Twenty-four hours later, mice were placed in a novel environment to remove contextual triggers and exposed to auditory cues from the previous day without footshock (**Figure 2B**). Memory precision was measured using per cent time freezing during presentation of neutral (CS-) and conditioned (CS+) auditory cues. We report that mice accurately learn this task and can readily distinguish between neutral and conditioned cues (**Supplementary file 1**).

Using this paradigm, we observed no effect of stress on freezing responses to the CS- presentations during conditioning (**Figure 2C**, grey area), ruling out ELS-induced hypersensitivity or hypervigilance to novel auditory cues. We also found that fear acquisition, evidenced by freezing to the CS+/US pairing, was unaffected by ELS (**Figure 2C**). In contrast, memory recall was affected with ELS mice exhibiting enhanced fear responses to neutral cues (CS-; **Figure 2D**, grey area), with no difference in freezing to conditioned cues (CS+; **Figure 2D**) compared to naïve mice. Additional analysis comparing average freezing responses to all tone presentations of CS- and CS+ supported these observations with no differences between naïve and ELS during conditioning (**Figure 2—figure supplement 2A**); while during memory recall ELS mice showed enhanced freezing to CS- presentations compared to naïve counterparts with no difference in freezing to CS+ (**Figure 2—figure supplement 2B**).

ELS-induced enhancement of fear responses to neutral cues is indicative of fear generalisation, a common stress-related behavioural phenotype conserved between rodents and humans (**Asok et al., 2018**; **Dymond et al., 2015**). This behavioural observation is more clearly depicted using a discrimination index (DI) that takes into account freezing responses to neutral and conditioned cues. Using this measure, DI > 0 reflects accurate discrimination, DI = 0 no discrimination, and DI <0 reflects a discrimination error, that is, higher freezing to the neutral than conditioned cue. Using this index, we confirm a decrease in fear discrimination in ELS mice compared to naïve mice (**Figure 2E**). Together these data reveal long-term behavioural changes with ELS enhancing affective responses resulting in fear generalisation in adult mice.

Investigation of potential sex effects revealed no differences between naïve male and female mice in this paradigm, with both sexes showing equal performance in learning (**Figure 2—figure supplement 1A**), memory (**Figure 2—figure supplement 1B**), and auditory discrimination (**Figure 2—figure supplement 1C**). In ELS mice, we report a sex-dependent effect on fear acquisition, with female mice exhibiting increased freezing responses to CS+/US pairing compared to males (**Figure 2—figure supplement 1D**). Despite this, both sexes show equal performance during memory recall and auditory discrimination (**Figure 2—figure supplement 1E and F**).

Auditory fear conditioning paradigms have been correlated with the potentiation of postsynaptic responses to cortical auditory afferents, in the lateral amygdala (**Kim and Cho, 2017**). As such, we set out to determine whether ELS impacts synaptic plasticity in cortico-amygdala circuits using acute brain slice electrophysiology and recorded field excitatory postsynaptic potentials (fEPSPs) by stimulating cortical inputs from the external capsule (**Figure 2F**). We report that ELS impaired long-term potentiation (LTP) of synaptic transmission in this cortico-amygdala circuit (**Figure 2 G and H**, **Figure 2—figure supplement 2**). This synaptic observation is consistent with findings using distinct stress paradigms

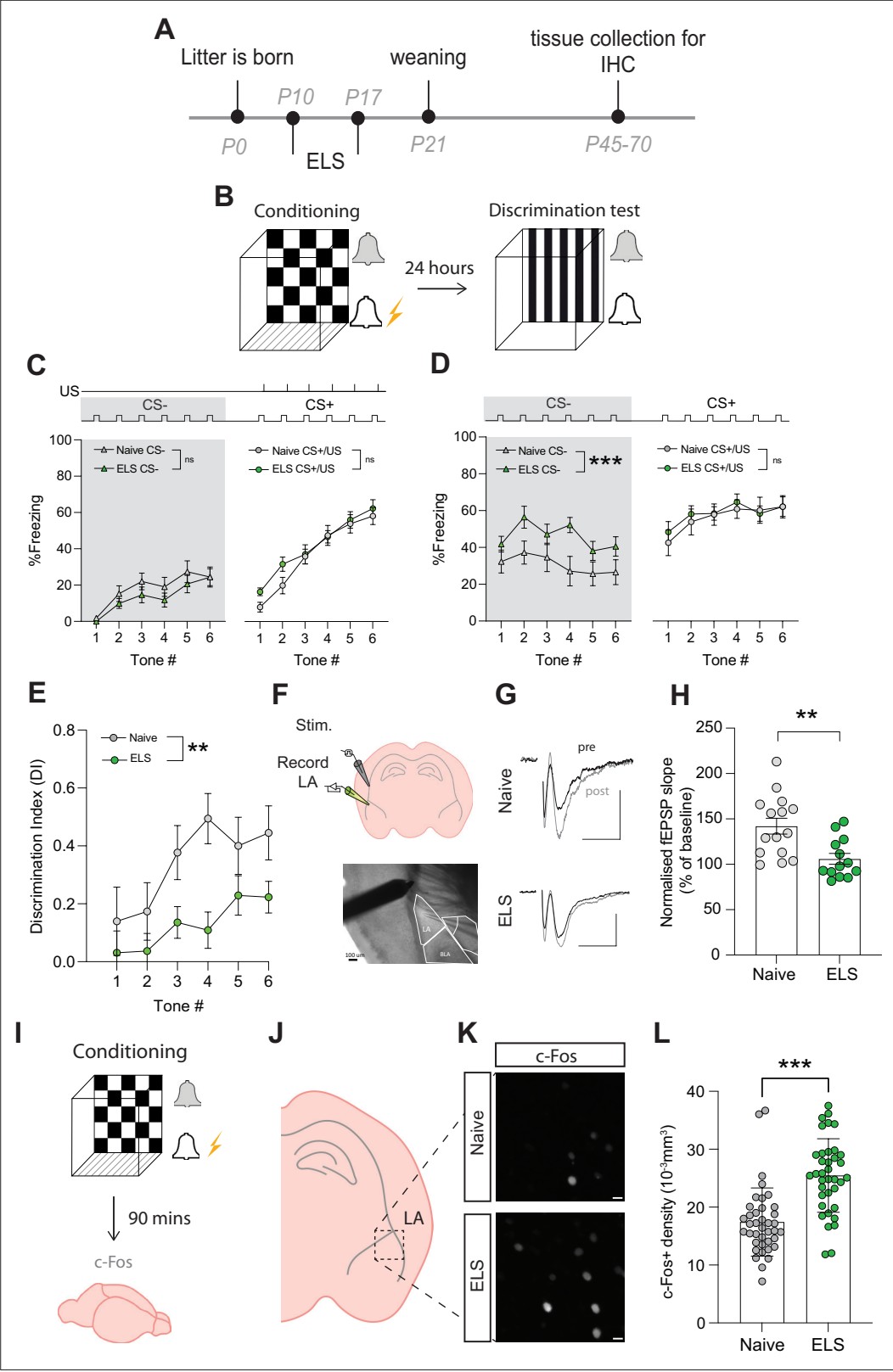

**Figure 2.** Early-life stress (ELS) alters fear memory, synaptic plasticity, and neural excitability. (**A**) Experimental timeline. (**B**) Timeline of auditory discriminative fear conditioning paradigm. (**C**) ELS did not affect % freezing to neutral and aversive auditory cues during learning. Naïve N = 20, ELS N = 17. (**D**) % time freezing during presentation of neutral auditory cue (CS-; p=0.0002) and aversive cue (CS+; p=0.99). Naïve N = 20, ELS N = 17.

*Figure 2 continued on next page*

*Figure 2 continued*

(**E**) Discrimination Index was impaired by ELS. p=0.004, naïve N = 20, ELS N = 17. (**F**) Schematic representation of electrode placement in slice electrophysiology experiment (top) with DIC image below. (**G**) Representative EPSP traces from naïve (top) and ELS (bottom) brain slices during baseline (black trace) and after LTP stimulation (grey trace). Scale bars = 0.5 mV and 10 ms. (**H**) LTP was impaired by ELS. p=0.002, naïve N = 15, ELS N = 13. (**I**) Diagram of procedure for c-Fos staining experiments. (**J**) Brain slices containing lateral amygdala (LA) were selected for c-Fos staining. (**L**) Region of interest extracted from full amygdala image. Scale bars = 10 μm. (**K**) c-Fos-positive cell density was increased by ELS. p<0.0001, naïve n = 39 slices N = 10 mice, ELS n = 37 slices N = 10 mice. See *Supplementary file 1* for detailed statistical summaries.

The online version of this article includes the following figure supplement(s) for figure 2:

**Figure supplement 1.** Effect of sex on amygdala-dependent learning and memory in naive and ELS mice.

**Figure supplement 2.** Sex-dependent effects of early-life-stress (ELS) on neuronal excitabiliy following fear conditioning.

**Figure supplement 3.** Astrocytic expression of c-Fos after fear conditioning.

---

(*Murphy-Royal et al., 2020*) and highlights the important influence of stress on synaptic function across various brain regions.

Fear learning and memory in the lateral amygdala have also been correlated with the allocation of a small number of neurons to a fear engram (*Josselyn et al., 2015*). These engrams are thought to be a possible neural substrate for memory, and their formation depends on neural excitability levels at the moment of acquisition (*Rashid et al., 2016*; *Ramsaran et al., 2023*). As both glucocorticoid injections or acute stress have been shown to increase engram size and enhance fear generalisation (*Lesuis et al., 2021*; *Lesuis et al., 2025*), we set out to determine whether ELS results in memory changes due to alterations in putative engram recruitment. We collected brain tissue 90 min after conditioning (*Figure 2I*) and carried out immunostaining for the immediate early gene c-Fos, a commonly used activity marker, in the lateral amygdala (*Figure 2J–L*, *Figure 2—figure supplement 2D and E*). Fear acquisition in ELS mice resulted in a significant increase in c-Fos density, suggestive of increased excitability levels. These results are consistent with reports of c-Fos density inversely correlating with memory accuracy (*Ramsaran et al., 2023*). Further investigation of potential sex differences revealed that while there were no differences in c-Fos densities between naïve female and male animals (*Figure 2—figure supplement 2F*), we observed a significantly higher c-Fos density in ELS male mice compared to females following fear acquisition (*Figure 2—figure supplement 2G*).

Recently, astrocytes have been reported to be recruited to fear engrams during a hippocampus-dependent learning and memory task (*Williamson et al., 2024*). As such, we quantified the percentage of astrocytes expressing the activity marker c-Fos following fear conditioning in mice and report similar values to those found in the hippocampus (*Figure 2—figure supplement 3A and B*) suggestive of modest recruitment of astrocytes in amygdala-dependent memory engrams.

## ELS induces persistent changes in astrocytes

To determine the influence of ELS on astrocytes in the lateral amygdala, we carried out immunostaining on brain slices taken from a separate cohort of naïve and ELS mice. We first investigated whether increased blood corticosterone affects glucocorticoid receptor (GR) expression and localisation in astrocytes. Using immunostaining, we confirmed that this hormone receptor is abundantly expressed in lateral amygdala astrocytes (*Figure 3B*). We further investigated the subcellular localisation of GR within astrocytes. When inactivated GRs are restricted to the cytosol, upon binding to agonists such as corticosterone, however, GRs can translocate to the nucleus to influence gene expression (*de Kloet et al., 2005*). Calculating a ratio of cytosolic vs nuclear GR in astrocytes as a proxy for activity, we found an increase in the nuclear fraction of GR in astrocytes following ELS (*Figure 3C*), suggestive of increased activation of glucocorticoid signalling in lateral amygdala astrocytes following ELS.

To assess putative changes in astrocyte structure, we investigated expression of the intermediate filament protein glial fibrillary acidic protein (GFAP; *Figure 3D and E*) and found a significant decrease in normalised GFAP fluorescence intensity in the lateral amygdala following ELS (*Figure 3F*). While functional consequences of GFAP changes are difficult to define, we interpret these modifications in GFAP expression to be indicative of an astrocytic response to stress. We next targeted astrocyte-specific proteins with known roles in influencing synaptic function including Cx43, a gap junction

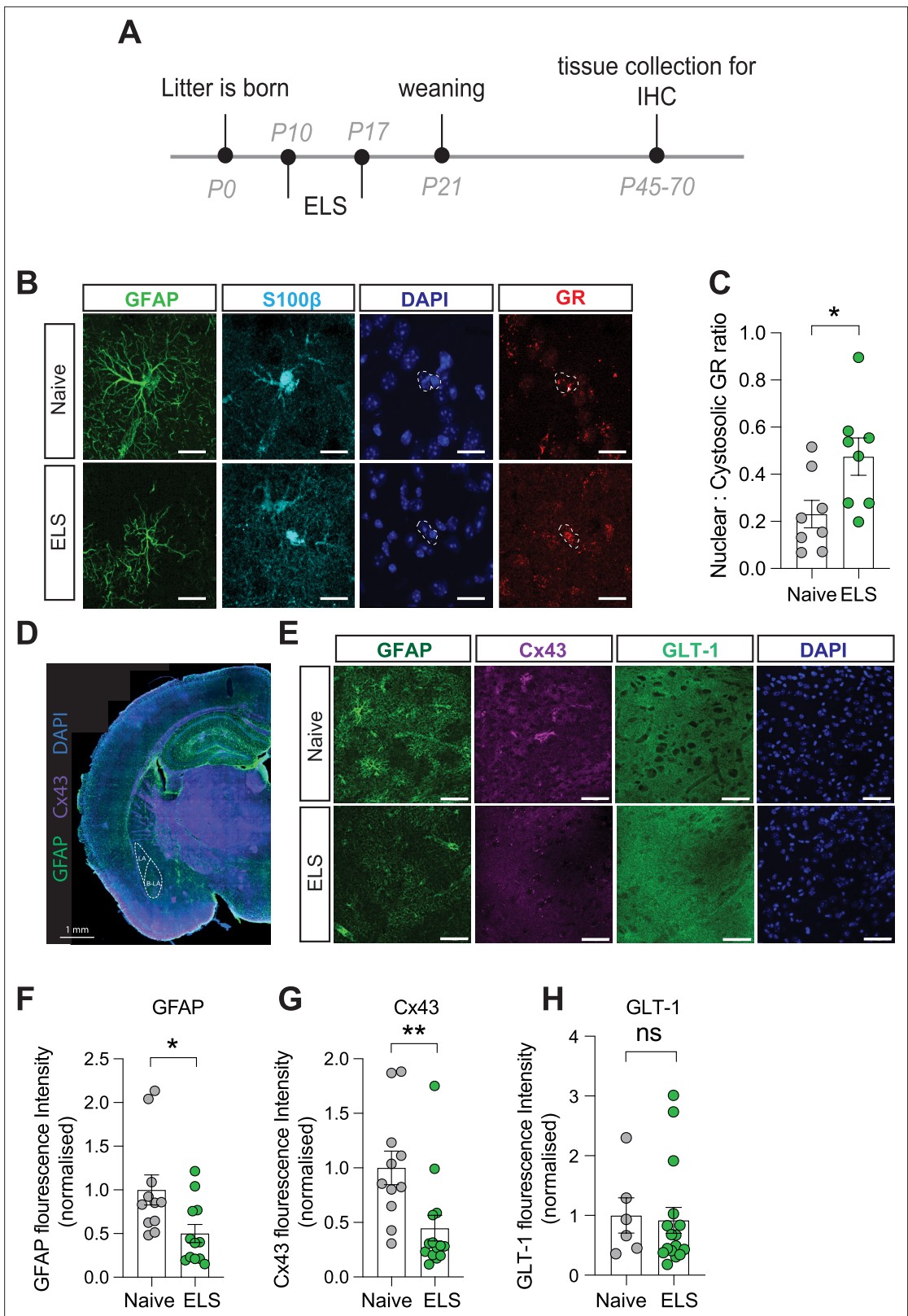

**Figure 3.** Long-term astrocyte dysfunction after early-life stress. (**A**) Experimental timeline. (**B**) Representative immunostaining of GFAP, S100β, GR, DAPI, and GR/DAPI merge in naïve and early-life stress (ELS) conditions. Scale bars = 20 μm. (**C**) Nuclear/cystolic GR ratio in astrocytes was increased after ELS. p=0.026, naïve N = 8, ELS N = 8. (**D**) Representative slide scan image of immunostaining for astrocyte proteins GFAP and Cx43. (**E**) Representative immunostaining of GFAP, Cx43, GLT-1, and DAPI in naïve and ELS conditions. Scale bars = 50 μm (**F**) GFAP staining was reduced in

*Figure 3 continued on next page*

*Figure 3 continued*

ELS mice. p=0.02, Naïve N = 11, ELS N = 12. (**G**) Cx43 staining was decreased in ELS mice. p=0.008, naïve N = 11, ELS = 14. (**H**) GLT-1 expression was unchanged by ELS. p=0.8, naïve N = 6, ELS N = 16. See **Supplementary file 1** for detailed statistical summaries.

The online version of this article includes the following figure supplement(s) for figure 3:

**Figure supplement 1.** Absence of sex-differences in GFAP and Cx43 in naïve and ELS mice.

protein that composes astrocyte networks (*Giaume et al., 2010*) and the glutamate transporter GLT-1, responsible for the efficient clearance of glutamate from the synaptic cleft (*Kruyer et al., 2022*; *Figure 3D and E*).

We observed a decrease in fluorescence intensity for the gap junction protein Cx43 (*Figure 3G*) while expression of the glutamate transporter GLT-1 remained unaffected by ELS (*Figure 3H*). We did not observe any significant sex differences in the expression of GFAP or Cx43 in control or ELS conditions, with similar expression levels in naïve and the same effect of stress on these proteins in both male and female mice (*Figure 3—figure supplement 1A and B*). These data suggest that astrocytes mount a specific response to ELS which could result in discrete changes in astrocyte network function such as the uncoupling of the astrocytic Cx43 gap junction networks in the lateral amygdala.

## ELS impairs astrocyte calcium activity

While the ELS-induced changes in astrocyte protein expression are indicative of functional modifications, this cannot be assumed and needs to be tested. We chose to assess potential stress-induced changes in astrocyte calcium activity employing the membrane-tethered calcium sensor lck-GCaMP6f. Lateral amygdala astrocytes were specifically targeted using local injection of viral constructs with GCaMP6f expression driven by the GfaABC1D promoter. Two-photon laser scanning microscopy was used in acute brain slices a minimum of 2 weeks following virus injection (*Figure 4A*), allowing for quantification of dynamic elevations in intracellular calcium in astrocytic processes in naïve (*Figure 4B and C*) and ELS mice (*Figure 4D and E*). Data was analysed using the AQUA2 pipeline, allowing for accurate and unbiased analysis of spatial and temporal dynamics of astrocyte calcium activity (*Mi et al., 2024*).

We report that ELS results in astrocyte calcium hypofunction, evidenced by decreased calcium event frequency (*Figure 4F and G*), amplitude (*Figure 4H and I*), and size of individual events (*Figure 4J and K*). Additionally, we observed a modest increase in rise time of calcium transients (*Figure 4L and M*) with no significant change in event duration (*Figure 4N and O*) or decay time (*Figure 4P and Q*). Together, these results are suggestive of astrocyte hypofunction following ELS.

Unexpectedly, our experiments revealed sexual dimorphism in astrocytic calcium dynamics in the lateral amygdala between naïve female and male mice. Specifically, we found that naïve female mice exhibited larger event amplitudes (*Figure 4—figure supplement 1A and B*), event area *Figure 4— figure supplement 1D and E*, and rise time (*Figure 4—figure supplement 1G and H*) compared to naïve male mice. Interestingly, all sex differences were suppressed by stress with ELS-induced decrease in astrocyte activity normalising all parameters between sexes (*Figure 4—figure supplement 1A, C, D, E, G, and I*).

No sex differences were observed for event frequency (*Figure 4—figure supplement 1J–L*), duration (*Figure 4—figure supplement 1M–O*), or decay time (*Figure 4—figure supplement 1P–R*) in any conditions.

## Astrocyte dysfunction alone phenocopies the effects of ELS on fear generalisation, synaptic plasticity, and excitability

To implicate astrocytes in the cellular and behavioural effects of ELS, we carried out a series of genetic loss-of-function experiments, specifically targeting astrocytes in the lateral amygdala using viral approaches (*Figure 5A–C*, *Figure 5—figure supplement 1A*). Based on the putative astrocyte network dysfunction we identified in fixed tissue (*Figure 3G*), we targeted astrocyte network function by overexpressing a dominant negative Cx43 (dnCx43). We have previously shown this manipulation to completely occlude functional coupling between neighbouring astrocytes by introducing a point mutation that blocks the pore of gap junction channels (dnCx43) (*Murphy-Royal et al., 2020*). Second, to mimic the impact of ELS on astrocyte calcium activity (*Figure 4*), we used a calcium extruder

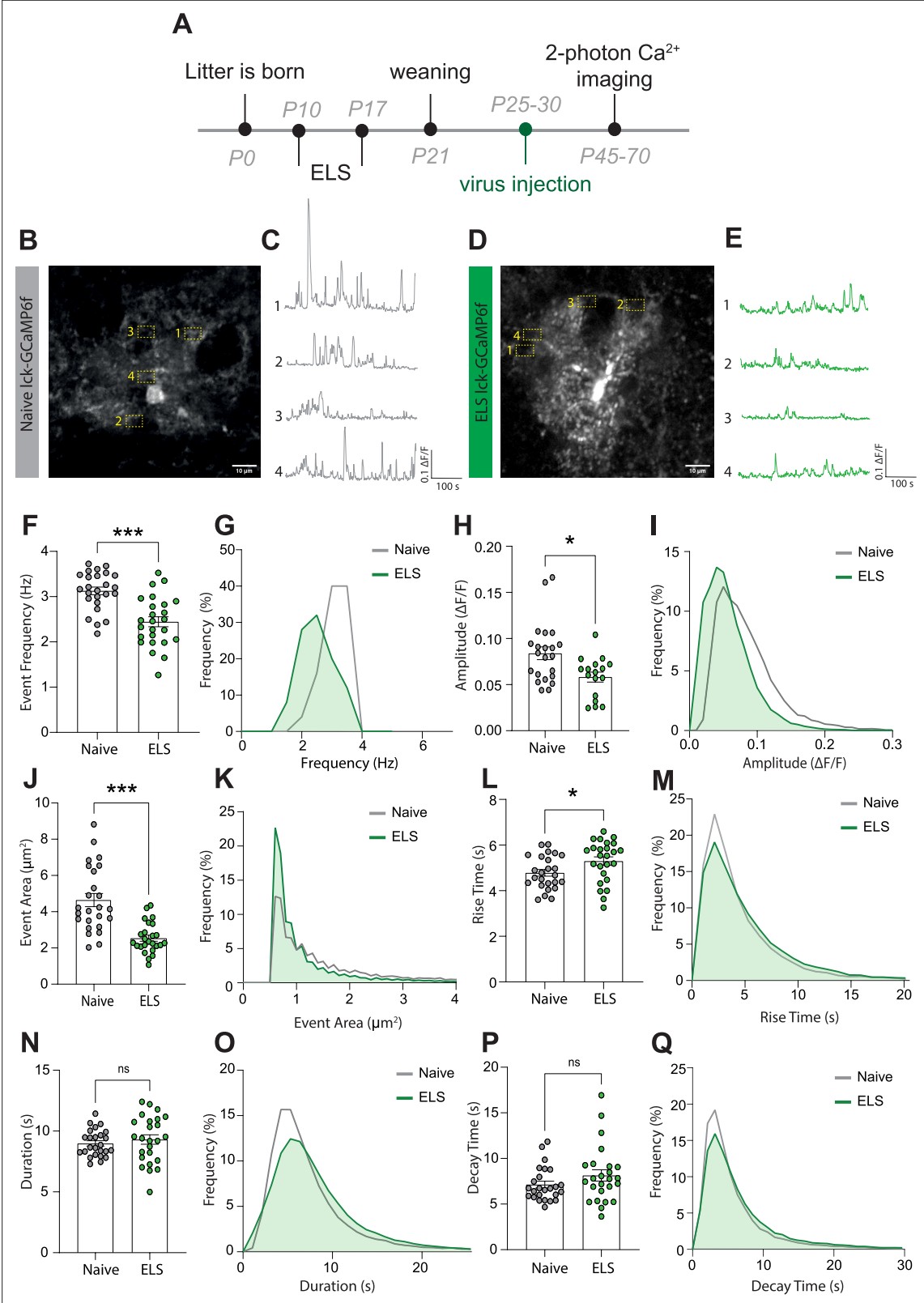

**Figure 4.** Early-life stress (ELS) induces a hypofunction in lateral amygdala astrocyte calcium signalling. (**A**) Experimental timeline including age of viral injection and experimental window for two-photon calcium imaging. (**B**) Representative image of a lck-GCaMP6f-expressing astrocyte in the lateral amygdala with four regions of interest from a naïve animal. Scale bar = 10 μm. (**C**) Representative calcium-signalling traces from four regions of interest shown in (**B**). (**D**) Representative image of a lck-GCaMP6f-expressing astrocyte in the lateral amygdala with four regions of interest from an ELS animal.

*Figure 4 continued on next page*

*Figure 4 continued*

Scale bar = 10 μm. (**E**) Representative calcium-signalling traces from four regions of interest shown in (**D**). (**F**) Frequency of calcium events was decreased lateral amygdala astrocytes after ELS, p<0.0001, naïve n = 25 astrocytes from N = 8 animals, ELS n = 25 astrocytes from N = 7 animals. (**G**) Frequency distribution of calcium event frequency from naïve (grey line) and ELS (green line) animals. (**H**) Amplitude of calcium events was decreased after ELS, p=0.0165, naïve n = 22 astrocytes from N = 8 animals, ELS n = 17 astrocytes from N = 7 animals. (**I**) Frequency distribution of calcium event amplitude from naïve (grey line) and ELS (green line) animals. (**J**) Area of calcium events was decreased after ELS, p<0.0001, naïve n = 25 astrocytes from N = 8 animals, ELS n = 25 astrocytes from N = 7 animals. (**K**) Frequency distribution of event area from naïve (grey line) and ELS (green line) animals. (**L**) Rise time of calcium events was increased following ELS, p=0.0311, naïve n = 25 astrocytes from N = 8 animals, ELS n = 25 astrocytes from N = 7 animals. (**M**) Frequency distribution of calcium event rise from naïve (grey line) and ELS (green line) animals. (**N**) Calcium event duration was unchanged by ELS, p=0.4740, naïve n = 25 astrocytes from N = 8 animals, ELS n = 25 astrocytes from N = 7 animals. (**O**) Frequency distribution of calcium event duration from naïve (grey line) and ELS (green line) animals. (**P**) Decay time of calcium events was unaltered by ELS, p=0.2164, naïve n = 25 astrocytes from N = 8 animals, ELS n = 25 astrocytes from N = 7 animals. (**Q**) Frequency distribution of calcium event decay time duration from naïve (grey line) and ELS (green line) animals.

The online version of this article includes the following figure supplement(s) for figure 4:

**Figure supplement 1.** Early-life-stress suppresses sexual dimorphism in lateral amygdala astrocyte calcium activity.

---

pump (hPMCA2/b, a.k.a. CalEx) (*Institoris et al., 2022*; *Yu et al., 2018*) that we validated to show that expression of this pump in lateral amygdala astrocytes (*Figure 5—figure supplement 2A–E*) effectively reduces frequency (*Figure 5—figure supplement 2F*) and amplitude (*Figure 5—figure supplement 2G*) of astrocyte calcium events without affecting event size, duration, rise time, or decay (*Figure 5—figure supplement 2H–K*). This manipulation closely mimics the effects of ELS on astrocyte calcium activity, allowing us to parse the precise contribution of astrocyte activity in behaviour.

Initially, we screened for changes in anxiety-like behaviours using the elevated plus maze and report no impact of lateral amygdala-specific astrocyte manipulation on the time spent in the open versus closed arms or in total distance travelled (*Figure 5—figure supplement 3A–C*) nor in time spent in the periphery versus the middle of the open-field maze or in total distance travelled (*Figure 5—figure supplement 3D–F*). Next, using auditory discriminative fear conditioning paradigm, we found that learning was unaffected by astrocyte dysfunction (*Figure 5D*, *Figure 5—figure supplement 4A*). We also report that similar to ELS, lateral amygdala astrocytic dysfunction does not lead to hypervigilance to neutral auditory cues (*Figure 5D*, grey area). During memory recall, however, we found that both astrocyte hypofunction manipulations enhanced fear responses to neutral cues, without affecting freezing to conditioned cues (*Figure 5E*, *Figure 5—figure supplement 4B*). This behavioural phenotype induced by astrocyte dysfunction alone, with either dnCx43 or CalEx, mimicked the effects of ELS on auditory fear discrimination, that is, enhanced fear generalisation. Accordingly, quantification of DI revealed an impairment in discriminative fear memory in both dnCx43 and CalEx conditions, compared to viral control (*Figure 5F*). This effect was dependent on the increase in freezing to the neutral cue with freezing responses to the aversive tone being unaltered by astrocytic dysfunction (*Figure 5—figure supplement 4B*). Taken together, these data directly implicate astrocytes in valence processing in the lateral amygdala and suggest a potential link between ELS and astrocyte dysfunction.

Investigation of potential sex differences in these genetic manipulations of astrocyte function revealed that the impact of CalEx on discrimination memory was more robust in female mice compared to males (*Figure 5—figure supplement 4C*). In the case of dnCx43, no differences were observed between females and males (*Figure 5—figure supplement 4D*).

Next, we set out to determine whether astrocyte dysfunction also mirrored the cellular and synaptic changes observed in ELS. Using acute brain slice electrophysiology, we report an occlusion of LTP in both CalEx and dnCx43 conditions (*Figure 5G*). In line with fear memory and synaptic plasticity changes, we also report that astrocyte dysfunction directly influences neural excitability with increased c-Fos density following fear conditioning in both CalEx and dnCx43 compared to control virus (*Figure 5H and L*). No sex differences were observed for c-Fos densities in eGFP- (*Figure 5—figure supplement 5A and D*) or CalEx-expressing mice (*Figure 5—figure supplement 5B and E*). dnCx43 expression, however, resulted in higher c-Fos density in female mice compared to male counterparts (*Figure 5—figure supplement 5F*).

Globally, these data highlight astrocytes as critical regulators of lateral amygdala function and output with astrocyte integrity essential for regulating synaptic plasticity, neural excitability, and of brain region-dependent behaviour (*Figure 6*).

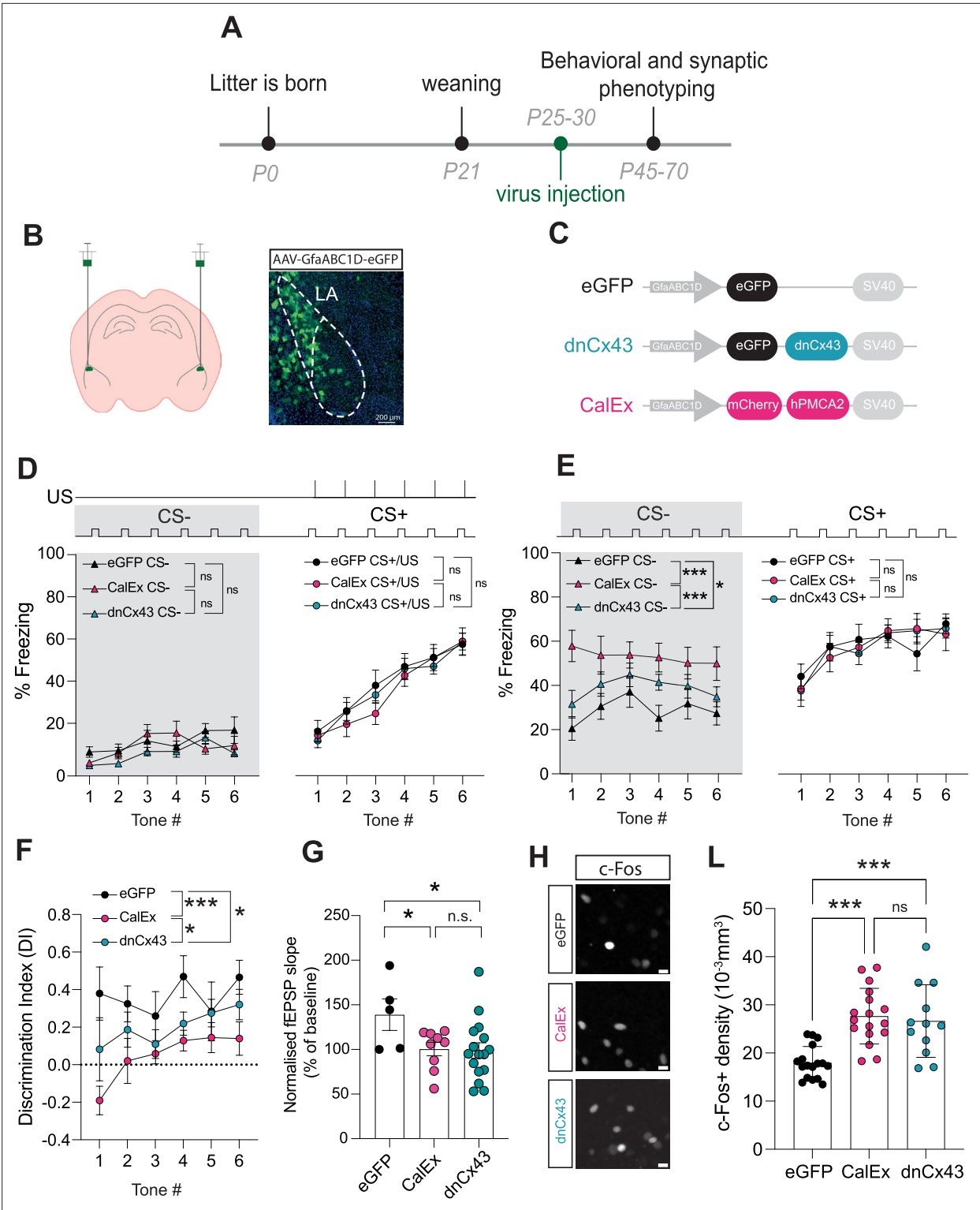

**Figure 5.** Astrocyte dysfunction mimics the effects of early-life stress (ELS) on behaviour, synapses, and neural excitability. (**A**) Experimental timeline. (**B**) Representative image showing localisation of viral vector in lateral amygdala. (**C**) Viral constructs used to manipulate astrocyte function. (**D**) Astrocyte dysfunction did not affect % freezing to neutral and aversive auditory cues during learning. eGFP N = 12, CalEx N = 11, dnCx43 N = 17 mice. (**E**) % time freezing during presentation of neutral auditory cue (CS-) was increased with astrocyte dysfunction (eGFP vs CalEx: p<0.0001; eGFP vs dnCx43: p=0.029; CalEx vs dnCx43: p=0.0005) with no impact on % freezing to aversive cue (CS+). eGFP N = 12, CalEx N = 11, dnCx43 N = 17 mice. (**F**) Discrimination Indexes were impaired by astrocyte dysfunction. eGFP vs CalEx: p<0.0001; eGFP vs dnCx43: p=0.013; CalEx vs dnCx43: p=0.03. eGFP N = 12, CalEx N

*Figure 5 continued on next page*

*Figure 5 continued*

= 11, dnCx43 N = 17 mice. (**G**) Astrocyte dysfunction occluded the induction of LTP. eGFP vs CalEx: p=0.04; eGFP vs dnCx43: p=0.02; CalEx vs dnCx43: p=0.88. eGFP N = 5, CalEx N = 9, dnCx43 N = 16. (**H**) Representative c-Fos staining in the lateral amygdala in control (eGFP), CalEx, and dnCx43 conditions. Scale bars = 10 μm. (**L**) c-Fos-positive cell density was increased with astrocyte dysfunction. eGFP vs CalEx: p<0.0001; eGFP vs dnCx43: p=0.0005; CalEx vs dnCx43: p=0.88. eGFP n = 17 slices N = 5 mice, CalEx n = 17 slices N = 5 mice, dnCx43 n = 12 slices N = 5 mice.

The online version of this article includes the following figure supplement(s) for figure 5:

**Figure supplement 1.** Viral targeting of lateral amygdala astrocytes.

**Figure supplement 2.** CalEx expression in lateral amygdala astrocytes suppresses calcium activity and closely mimics the effects of ELS on calcium activity.

**Figure supplement 3.** Impact of genetic manipulation of lateral amygdala astrocyte function on anxiety-related behaviours.

**Figure supplement 4.** Expression of CalEx, but not dnCx43, in lateral amygdala astrocytes results in sexually dimorphic effects on discrimination.

**Figure supplement 5.** Expression of dnCx43, but not CalEx, in lateral amygdala astrocytes influences neuronal excitability in sex-specific manner following fear conditioning.

## Discussion

In this study, we investigated the potential roles of astrocytes in cellular and behavioural dysfunctions associated with ELS. Using a rodent model of ELS that employs maternal separation and limited bedding, we report increased blood glucocorticoid levels associated with heightened anxiety-like behaviours. These endocrine and behavioural changes were accompanied by impaired discriminative

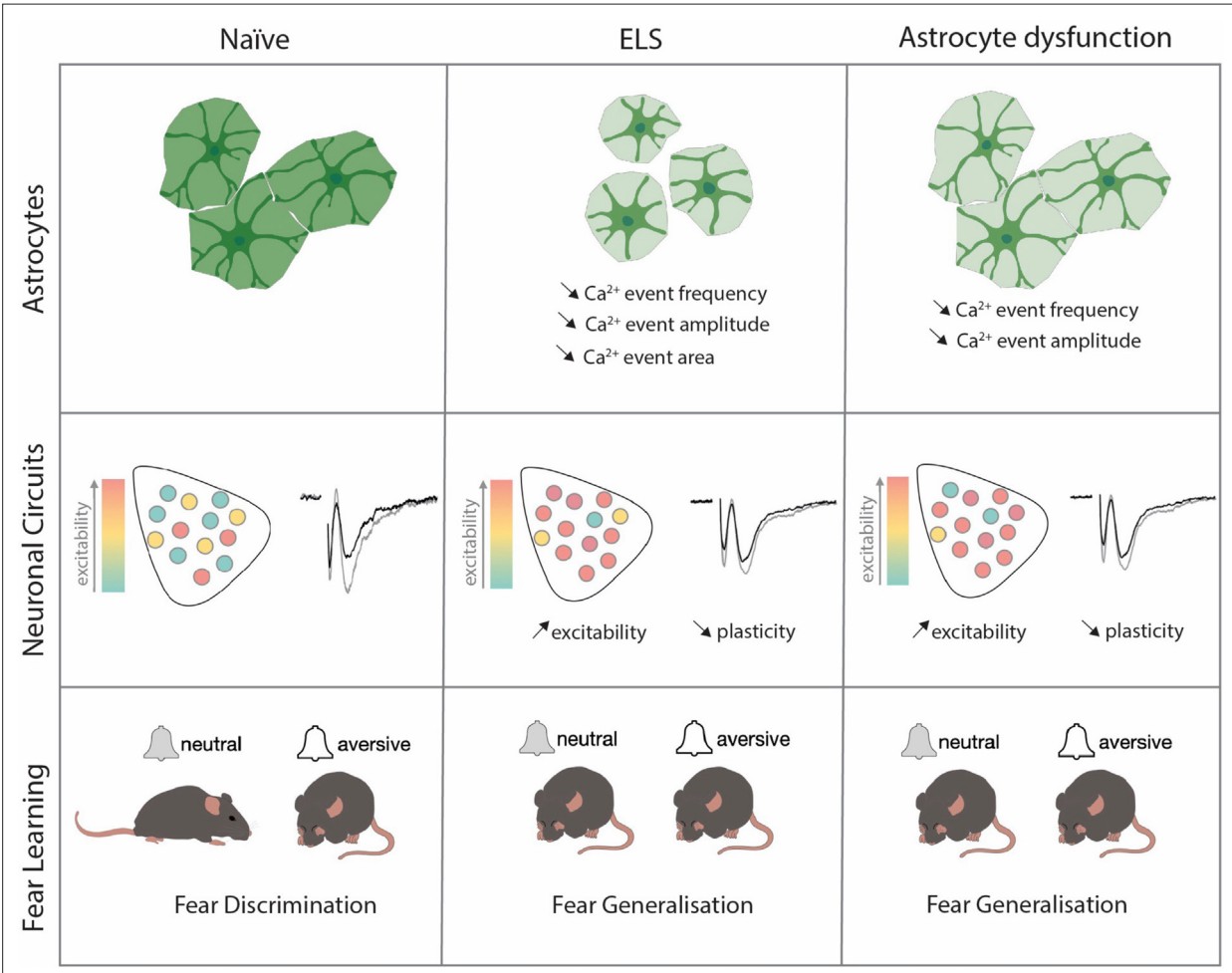

**Figure 6.** The effects of early-life-stress on astrocytic function, neuronal excitability, and fear memory are fully recapitulated by viral manipulations of lateral amygdala astrocytes.

fear memory in an amygdala-dependent auditory discrimination task. More precisely, we observed that ELS mice displayed increased fear responses to neutral cues with no change in freezing responses to conditioned stimuli. These behavioural changes were further associated with alterations on the cellular scale including synaptic plasticity and neuronal excitability. Investigating contributions of non-neuronal cells, we found robust changes in lateral amygdala astrocytes following ELS comprising increased GR translocation to the nucleus and decreased expression of key astrocyte network proteins. Functionally, we observed astrocyte hypoactivity with decreased frequency, amplitude, and size of calcium events following ELS.

To link astrocyte dysfunction with cellular, synaptic, and behavioural changes, we genetically targeted lateral amygdala astrocytes to mimic the effects we found in ELS. Astrocyte dysfunction alone, either decreasing gap junction coupling or reducing astrocyte calcium activity, phenocopied the effects of ELS across excitability, synaptic, and behavioural scales. Astrocytes have long been shown to be critical for memory formation, with the vast majority of studies reporting that astrocyte dysfunction impairs memory across various brain regions (*Robin et al., 2018*; *Yu et al., 2018*; *Depaauw-Holt et al., 2024*; *Sun et al., 2012*). Here, we show that astrocyte dysfunction in the lateral amygdala effectively impairs discriminative fear memory evidenced by fear generalisation, that is, the inappropriate expression of fear towards stimuli of neutral valence. These data reveal that astrocytes are critical for the appropriate behavioural responses to emotionally salient cues and provide evidence supporting the recently proposed contextual guidance theory of astrocyte function in the brain (*Murphy-Royal et al., 2023*).

Using an ELS paradigm that coincides with peak astrocyte postnatal development (*Rurak et al., 2022*; *Farhy Tselnicker and Allen, 2018*), we found a latent increase in circulating glucocorticoids after ELS with peak levels appearing during late adolescence and subsiding towards adulthood. While the lack of endocrine response to stress during the stress paradigm might appear paradoxical, this ELS paradigm overlaps with the stress hyporesponsive period, a time window during which stressors fail to elicit increases in blood corticosterone levels (*Schmidt et al., 2003*; *Sapolsky and Meaney, 1986*; *Schmidt, 2019*; *Vázquez, 1998*). Indeed, similar to our findings, others have reported latent increases in blood corticosterone levels following ELS that were associated with increased anxiety-like behaviour (*Sharma et al., 2022*) and fear generalisation (*Elliott and Richardson, 2019*) later in life. Increased levels of blood glucocorticoids had a direct impact on astrocytes with increased translocation of GR from the cytosol to the nucleus of astrocytes following ELS, indicative of a higher levels of glucocorticoid signalling (*de Kloet et al., 2005*). While the effects of ELS on blood glucocorticoids were transient, this could have long-term consequences on protein expression levels, resulting in changes in stress susceptibility and behavioural deficits. The primary behavioural phenotype revealed by this work, that is, fear generalisation, is a common trait observed in anxiety disorders (*Dymond et al., 2015*; *Lissek and van Meurs, 2015*; *Lis et al., 2020*). As such, these findings provide further insight into the neurobiological mechanisms of stress on endocrine, cellular, and behavioural scales.

Our work builds upon a growing literature that is interested in the potential roles of astrocytes in ELS (*Vivi and Di Benedetto, 2024*; *Abbink et al., 2019*; *Gunn et al., 2013*). Here, we identified specific astrocytic impairments induced by ELS and followed up with precise genetic manipulation of these astrocyte functions in the lateral amygdala. These astrocyte-specific manipulations replicate cellular and fear generalisation phenotype observed following ELS. We did not, however, observe an anxiety-like phenotype using elevated plus maze and open field with local astrocyte manipulation which could be interpreted as a lack of amygdala astrocyte involvement in these discrete behaviours.

We found that both ELS and astrocyte dysfunction enhance neuronal excitability, assessed by local c-Fos staining in the lateral amygdala following auditory discriminative fear conditioning. One interpretation of these data is that astrocytes might tune engram formation, with astrocyte dysfunction, genetically or after stress, increasing c-Fos expression and a loss of specificity of the memory trace. In support this notion, evidence of a cellular mechanism whereby c-Fos levels inversely correlate with memory accuracy has been recently shown in hippocampal circuits (*Asok et al., 2018*). In addition, early-life experience strongly shapes episodic memory development, with stress delaying maturation of memory processes (*Ramsaran et al., 2024*). Early-life stress has even been shown to activate specific neuronal ensembles that contribute to stress hypersensitivity (*Balouek et al., 2023*). Further support comes from recent demonstration of astrocyte-engram neuron interactions with astrocytes potentially forming part of engram stabilisation or even direct recruitment of astrocytes into engrams

(*Williamson et al., 2024*; *Kim et al., 2023*). Our data are in line with previous reports that have demonstrated clear roles for astrocytes in memory formation and consolidation (*Martin-Fernandez et al., 2017*; *Adamsky et al., 2018*; *Kol et al., 2020*) and reveal that astrocytes participate in the neurobiological mechanisms by which stress influences emotional memory.

Our work revealed unexpected sex differences in calcium signalling in lateral amygdala astrocytes. We found astrocyte calcium events were larger in amplitude and size in female mice compared to males. This higher basal activity could explain why CalEx induced a more profound impairment in fear discrimination in female mice. Interestingly, these sex differences are eliminated by stress inducing a homogenous calcium signature between both sexes. Sex differences in astrocytic development and function is starting to gain increased attention (*Depaauw-Holt et al., 2024*; *Rurak et al., 2022*; *González-Vila et al., 2023*), with much more to be learned regarding the potentially divergent functions of astrocytes in the female and male brain.

In sum, our data shed new light on the role of astrocytes as central regulators of amygdala-dependent behaviour, synaptic function, and excitability. In addition, our work suggests a key role of astrocytes in mediating stress-induced behavioural and cellular impairments, supporting the notion that astrocytic disruption is not simply downstream of neuronal dysfunction. These ideas are forcing a change in perception of how brain disorders develop, moving away from a neuro-centric cascade and placing astrocytes in the spotlight (*Murphy-Royal et al., 2023*; *Vivi and Di Benedetto, 2024*; *Seifert et al., 2006*; *Lee et al., 2022*; *Endo et al., 2022*; *Khakh and Goldman, 2023*). Collectively, these data support the hypothesis that astrocytes are key to execute the appropriate behavioural response to emotionally salient cues and underscore the potential of astrocytes as therapeutic targets in stress-related psychiatric conditions.

## Materials and methods

### Animals
All experiments were performed in accordance with the guidelines for maintenance and care of animals of the Canadian Council on Animal Care and approved by the Institutional Committee for the Protection of Animals at the Centre Hospitalier de l'Université de Montréal, protocol number CM20031CMRs. Both male and female C57BL/6J mice (Jax #000664) were used in the study, housed on a 12 hr:12 hr light:dark cycle (lights on at 06:30) with ad libitum access to food and water.

### ELS protocol
ELS was performed as previously described (*Peña et al., 2017*).C57BL/6J pups were separated from their mothers for 4 hr per day (ZT2-ZT6) for 7 days between ages P10 and P17. During separation, pups and mothers were placed into new cages with 60% less bedding. Bedding was weighed on P10 and divided 33% into each cage (home cage, pup separation cage, and mother separation cage). Separated pups and mothers had ad libitum access to food and water and pup cages were placed on a water heating pad maintained at 34°C. After the final day of separation (P17), bedding was returned to the original amount and pups were housed with their mothers until weaning at P21.

### Behavioural assays
#### Auditory discriminative fear conditioning
Mice between ages P45-70 were used. Four days prior to behavioural testing, mice were single housed in individual cages. Two days later, mice were handled for approximately 10 min by hand and mobilisation tube. On the first day of ADFC, single housed mice were left for 1 hr in behavioural room to habituate prior to the start of the conditioning assay.

#### Conditioning
Mice were first exposed to a neutral stimulus (CS-; white noise, 20 s) repeated six times. After a 2-min interval, mice were exposed to a tone which becomes the conditioned stimulus (CS+; 12 kHz tone, 20 s) as it co-terminates with a mild foot shock (US; 0.5 mA, 2 s) delivered in the last 2 s of tone presentation (repeated six times). Conditioning was carried out in chamber with checkered walls that was cleaned with 70% ethanol between animals.

## Memory recall

Then, 24 hr after conditioning, mice were exposed to CS- and CS+ (counterbalanced) in a novel environment characterised by stripped walls and a white floor covering, cleaned with 0.5% hydroperoxide between animals. Fear learning was quantified by the percentage of time spent freezing to CS- and CS+/US during the conditioning phase. Fear memory was quantified by the percentage time freezing to CS- or CS+ following presentation in the memory-recall phase. A DI was calculated comparing fear responses to CS+ and CS- using the following formula:

$$Discrimination\ Index\,(DI) = \frac{(CS^+ Freezing) - (CS^- Freezing)}{(CS^+ Freezing) + (CS^- Freezing)}$$

## Open-field task

This task comprised of a total of 5 min and during that time the mice were left to freely explore the OFT apparatus, a white plexiglass box measuring 38 × 38 cm. During the task, mice were filmed, and the time spent in distinct parts of the box, that is, centre and periphery were quantified using AnyMaze software (Stoelting). The total time spent in the centre was used as a quantification of anxiety-like behaviour, where more time in the centre represented less anxiety-like behaviour and the total distance travelled was measured to quantify locomotor changes.

## Elevated plus maze

This task comprised of a total of 5 min during which mice were removed from their cages and placed at the centre of the four arms of the maze, facing an open arm. Mice were video recorded, and these videos were analysed using AnyMaze software (Stoelting) which quantified the time spent (s) in closed arms or open arms during the 5 min testing period. Anxiety-like behaviour was quantified by ratio of time spent in open vs closed arms.

## Acute brain slice preparation

Coronal slices containing lateral amygdala were obtained from mice between P45 and 70. Animals were anaesthetised with isoflurane and the brain was rapidly excised and placed in ice-cold choline-based cutting solution saturated with 95% $O_2$ and 5% $CO_2$ containing the following (in mM): 120 choline chloride, 3 KCl, 1.25 $NaH_2PO_4$, 26 $NaHCO_3$, 8 $MgCl_2$, 20 glucose, pH 7.2, and 295 mOsmol. Slices (300 μm thick) were cut on a vibratome (Leica VT1200, Nussloch, Germany). Slices were transferred to oxygenated artificial CSF (ACSF) at 32 ± 0.5°C for 30 min containing the following (in mM): 130 NaCl, 2.5 KCl, 1.25 $NaH_2PO_4$, 26 $NaHCO_3$, 10 glucose, 1.3 $MgCl_2$, 2 $CaCl_2$, pH 7.3–7.4, and 300–305 mOsmol, then allowed to recover for at least 30 min before recordings in oxygenated ACSF at room temperature (RT). For experiments, the slices were transferred to a recording chamber where they were perfused (2 ml/min) with ACSF at 32°C for the course of the experiment. Slices were used for a maximum of 6 hr after cutting.

## Electrophysiology

Extracellular fEPSPs were recorded in current-clamp mode with a pipette filled with ACSF placed in the lateral part of the amygdala. Monosynaptic EPSPs were elicited by stimulation of cortical afferences with a tungsten concentric bipolar electrode (World Precision Instruments). LTP was induced using a theta burst stimulation protocol consisting of four pulses at 100 Hz repeated at 5 Hz intervals for 10 s. Signals were amplified using a Multiclamp 700B amplifier (Molecular Devices) and digitised using the Digidata 1440 (Molecular Devices). Data were recorded (pClamp 10.2, Molecular Devices) for offline analysis. The magnitude of LTP was quantified by normalising and averaging the slope of fEPSPs following LTP induction relative to the last 5 min of baseline.

## Two-photon calcium imaging and analysis

Fluorescence calcium imaging was performed on a custom two-photon laser-scanning microscope (Scientifica). The microscope was equipped with a Ti:Sapph laser (Ultra II, Coherent), a green bandpass emission filter (525–40 nm), an orange/red bandpass emission filter (605–70 nm), and associated photomultiplier tubes (GaAsP, Hamamatsu). Time-series images were acquired through bidirectional laser scanning (920 nm excitation laser, 86.85 μm × 86.85 μm, 512 × 512 pixels at 1 Hz) at a single

focal plane incorporating the entirety of the imaged astrocyte ramifications. All images were acquired using Scanimage (MBF Bioscience). Acute brain slices (350 μm) containing the lateral amygdala were obtained from mice (ages between P45 and 70) injected with the viral construct AAV2/5-gfaABC1D-lck-GCaMP6f 3 weeks prior. Animals were anaesthetised with isoflurane and the brain was rapidly excised and placed in ice-cold NMDG-based cutting solution saturated with 95% $O_2$ and 5% $CO_2$ containing the following (in mM): 119.9 NMDG, 2.5 KCl, 25, 1 $CaCl_2$, 1.4 $NaH_2PO_4$, and 20 D-glucose saturated with 95% $O_2$ and 5% $CO_2$. Slices (350 μm thick) were cut on a vibratome (Leica VT1200, Nussloch, Germany). Slices were transferred to oxygenated NMDG cutting solution ACSF at 33 ± 0.5°C for 15 min followed by a 1 hr recovery period in ACSF at RT containing the following (in mM): 130 NaCl, 2.5 KCl, 1.25 $NaH_2PO_4$, 26 $NaHCO_3$, 2.5 glucose, 1.3 $MgCl_2$, 2 $CaCl_2$, pH 7.3–7.4, and 300–305 mOsmol. All time-series images were acquired in a recording chamber constantly perfused with ACSF as described above.

For unbiased, automatic calcium event detection, TIFF files of time-lapse images were processed with the AQUA2 pipeline in MATLAB. Parameters of interest (event frequency, amplitude, area, duration, rise time, and decay time) were directly extracted from the AQUA2 output file. For statistical testing, averages of all events from a single astrocyte were calculated for each parameter. In parallel, to better represent the diversity in calcium events, frequency distributions plots were created considering every detected event for all our experimental conditions.

## Stereotaxic surgery and AAV delivery

P25-35 mice were given a subcutaneous (sc) injection of carprofen (20 mg/kg) 1 hr prior surgery. Animals were then deeply anaesthetised (2% isoflurane) before placing them into a stereotaxic frame (Kopf Instruments). Surgical site was infiltrated with bupivacaine/lidocaine (2 mg/kg sc) 15 min prior surgery. Bilateral injections of the following viral constructs produced at the Canadian Neurophotonics Viral Vector Core:AAV2/5-gfaABC1D-lck-GCamP6f (9.6 × 10$^{12}$ GC/ml), AAV2/5-gfaABC1D-eGFP (1.2–1.5 × 10$^{11–12}$ GC/ml), AAV2/5-gfaABC1D-dnCx43-eGFP (1.2–1.7 × 10$^{12}$ GC/ml), and AAV2/5-gfaABC1D-hPMCAw/b-mCherry (1.2 × 10$^{12}$) were delivered to the LA (coordinates relative to bregma: –1.4 mm AP, ±3.4 mm ML and –5 mm DV) of mice of either sex. Viral solutions were injected at a rate of 200 nl/min for a total volume of 650 nl per hemisphere using a 10 μl Hamilton syringe (Harvard apparatus) and a microinjection syringe pump (UMP3T-2 World Precision Instruments). Animals received 250 μl of saline for post-surgery hydration.

## Brain processing

Mice were anaesthetised in an enclosed chamber filled with isoflurane (5% for induction, 3–4% for maintenance, v/v) until loss of toe pinch reflex. A transcardial perfusion was used to ensure uniform preservation of tissue and brains were fixed in 4% paraformaldehyde for 24 hr at 4°C. The fixed brains were then placed in falcon tubes containing 30% sucrose solution for a minimum of 2 days. Brains were embedded in optimal cutting temperature compound and flash frozen in 2-methylbutane between –40 and –50°C and then stored at –80 until cryosectioning. Brains were cut at 30 um thickness at –20°C using a Leica CM3050S Cryostat. Slices were stored in cryoprotector solution at 4°C until immunohistochemistry.

For c-Fos labelling experiments, mice were anaesthetised and then perfused 90 min after the end of fear conditioning. The whole lateral amygdala was sectioned in 30-um-thick brain slices, and every fourth brain slice was selected for immunohistochemistry.

## Immunohistochemistry

Free-floating brain slices were washed three times in 1× PBS for 15 min to remove cryoprotector solution. Slices were permeabilised in block-perm solution (3% bovine serum albumin, 0.5% Triton$^{10\%}$ in PBS) for a duration of 1 hr. Slices were incubated with antibodies for 24 hr at 4°C. Slices were incubated with primary antibodies; [rabbit] anti-S100β (1:1000, Abcam, ab52642), [mouse] anti-GR (1:500, Thermo Fisher, MA1-510), [chicken] anti-GFAP (1:1000, Thermo Fisher, PA1-10004), [mouse] anti-Cx43 (1:1000 Thermo Fisher, 3D8A5), [rabbit] anti-GLT-1 (EAAT2) (1:1000, NovusBio, NBP1-20136), and [rabbit] anti-c-Fos (1:2000, Synaptic Systems, Cat# 226008). After primary antibody incubation, slices were washed three times in 1× PBS for 15 min to remove non-specific binding. Slices were incubated with secondary antibodies in a DAPI solution (1:10,000) at RT in aluminium foil for 1 hr. Slices were

incubated with secondary antibodies; [goat] anti-rabbit Alexa 488 (1:1000, Jackson ImmunoResearch, 111-545-144), [goat] anti-mouse Alexa 647 (1:1000, Thermo Fisher, A32728), and [goat] anti-chicken Alexa 568 (1:1000, Thermo Fisher, A11041). After secondary antibody incubation, slices were washed again three times in 1× PBS for 15 min before being mounted onto Fisherbrand microscope slides using ProLong Glass Antifade mountant (P36982).

## Microscopy

Slices were imaged using a Leica TCS SP5 laser scanning confocal microscope with oil immersion Plan-Apochromat ×63 objective 1.4 NA or using a Zeiss Observer Z1 spinning disk confocal microscope/ TIRF with a ×20 objective 1.8 NA. 16-bit images of 246 × 246 um areas were acquired at 400 Hz (frame size (x*y); 1024 × 1024, pixel size; 250 nm). 25–30 um Z-stacks were acquired with a step size of 0.5 um. 15 um max intensity z-projections of the lateral amygdala were analysed using ImageJ (Fiji) to obtain A.U. fluorescence intensity measures of secondary antibodies attached to primaries with specificity to epitopes; GFAP, Cx43, GLT-1, and GR. We applied a fluorescence threshold for GFAP, Cx43, and GLT-1 fluorescence and measured integrated density A.U. (thresholds; Mean, Otsu, Otsu, respectively). For nuclear and cytosolic GR measures, astrocyte (S100$\beta$+DAPI) nuclei were thresholded and used as regions of interest (ROIs) for nuclear measures of GR fluorescence (S100$\beta^+$ + DAPI$^+$). The same nuclear DAPI ROI was cleared from the previously thresholded s100b regions to obtain cyto-solic astrocytic GR (S100$\beta^+$ - DAPI$^+$). Integrated density A.U. of GR fluorescence was used to calculate nuclear:cytosolic GR ratios.

For c-Fos cell counting in the lateral amygdala, 16-bit images containing the lateral and basal amygdala nuclei were acquired. 20–30 Z-stacks were acquired with a step size of 1 um. For c-Fos particle counting analysis, a ROI was used to isolate the lateral amygdala and then 8-bit Z-stacks of the c-Fos channel were analysed semi-automatically by using the FIJI plugin Quanty-cfos (**Beretta et al., 2023**). In brief, a threshold for area and fluorescence intensity was set manually to identify ROIs around all c-Fos-positive nuclei. These ROIs were then manually inspected to remove false positives and negatives found by the plug-in. c-Fos particle counts were then normalised by the volume of the Z-stack to calculate the c-Fos density. In order to calculate the volume, the area of the lateral amyg-dala was multiplied by the number of Z-stacks.

## Blood collection and corticosterone analysis

Trunk blood was collected via decapitation at ZT2 (08:30). To minimise handling-induced elevations in corticosterone, mice were housed with clear plastic tubes used to move individual mice into an enclosed chamber filled with 5% isoflurane for 2 min. Mice were placed in enclosed isoflurane until loss of toe pinch reflect (maximum elapsed time of 2 min). Mice were decapitated and trunk blood was collected into BD 365963 Microtainer Capillary Blood Collection Tubes and placed onto ice. Blood was centrifuged for 5 min at 4°C at 5000 rpm. Serum was aliquoted and stored at –80°C. Corticoste-rone measurements were obtained using an ENZO ELISA kit (ADI-900-097).

## Statistical analyses

Detailed statistical analyses are provided in **Supplementary file 1**. Results are presented as mean ± SEM. Data with one variable were analysed with the two-tailed Student's *t*-test or Mann–Whitney test. Data with more than two conditions were first screened for a Gaussian distribution with Kolmog-orov–Smirnov test followed by analysis either with one-way/repeated-measures ANOVA or Kruskal–Wallis/Friedman test when needed and Tukey's multiple-comparison parametric *post hoc* test (data with Gaussian distribution) or by a Dunn`s multiple-comparison non-parametric *post hoc* test (data with non-Gaussian distribution). Auditory discriminative fear conditioning data was analysed using a two-way ANOVA with appropriate post hoc tests. Open-field and elevated plus maze data was analysed using one-way ANOVA with appropriate post hoc tests. Graphic significance levels were *$p<0.05$; **$p<0.01$, and ***$p<0.001$. All data were analysed using GraphPad Prism software (version 9, GraphPad, USA).

## Acknowledgements

We thank Rosemary Bagot (McGill University) for critical input on this study, Thierry Alquier and Stephanie Fulton (Université de Montréal) for their input throughout, Aurélie Cleret-Buhot (CRCHUM cellular imaging core) for microscopy training, and the staff of the animal facility at the CRCHUM. This work was funded by the Canadian Institutes of Health Research Project Grant (478629), NSERC Discovery Grant (RGPIN-2021-03211), Fonds de Recherche du Québec – Santé (FRQS; 296562 and 309889), Brain and Behaviour Research Foundation Young Investigator award (NARSAD; 28589), CHUM Foundation, Fondation Courtois, and the Réseau Québécois sur le Suicide les troubles de l'humeur et les troubles Associées (RQSHA) grant to CM-R. LD-H was supported by a doctoral fellowship from the Fonds de Recherche du Québec. IIA, JV, and BR were supported by a Canada Graduate Scholarship Master's award, and Recruitment Fellowships from Neuroscience Dept. Université de Montréal. CM-R was supported by a Junior 1 Chercheur-Boursier salary award from FRQS.

## Additional information

### Funding

| Funder | Grant reference number | Author |
| --- | --- | --- |
| Canadian Institutes of Health Research | 478629 | Ciaran Murphy-Royal |
| Natural Sciences and Engineering Research Council of Canada | RGPIN-2021-03211 | Ciaran Murphy-Royal |
| Fonds de Recherche du Québec - Santé | 296562 | Ciaran Murphy-Royal |
| Fonds de Recherche du Québec - Santé | 309889 | Ciaran Murphy-Royal |
| Fondation du CHUM | | Ciaran Murphy-Royal |
| Fondation Courtois | | Ciaran Murphy-Royal |
| Réseau Québecois sur le Suicide, les troubles de l'humeur, et les troubles associées | | Ciaran Murphy-Royal |
| Fonds de Recherche du Québec - Santé | Doctoral Scholarship | Lewis R Depaauw-Holt |
| Canadian Institutes of Health Research | Master's Scholarships | Ifeoluwa I Adedipe Juliette Vaugeois Benjamin Rogers |
| Brain and Behavior Research Foundation | Young Investigator NARSAD; 28589 | Ciaran Murphy-Royal |

The funders had no role in study design, data collection and interpretation, or the decision to submit the work for publication.

### Author contributions

Mathias Guayasamin, Lewis R Depaauw-Holt, Conceptualization, Data curation, Formal analysis, Validation, Investigation, Methodology, Writing – original draft, Writing – review and editing; Ifeoluwa I Adedipe, Conceptualization, Data curation, Formal analysis, Investigation, Methodology, Writing – original draft; Ossama Ghenissa, Data curation, Formal analysis, Writing – review and editing; Juliette Vaugeois, Manon Duquenne, Data curation, Formal analysis; Benjamin Rogers, Jade Latraverse-Arquilla, Data curation; Sarah Peyrard, Resources, Methodology, Project administration; Anthony Bosson, Conceptualization, Data curation, Formal analysis, Writing – original draft, Writing – review and editing; Ciaran Murphy-Royal, Conceptualization, Resources, Data curation, Formal analysis, Supervision, Funding acquisition, Validation, Investigation, Methodology, Writing – original draft, Project administration, Writing – review and editing

## Author ORCIDs
Mathias Guayasamin https://orcid.org/0009-0008-6099-930X
Lewis R Depaauw-Holt https://orcid.org/0009-0001-3797-447X
Ciaran Murphy-Royal https://orcid.org/0000-0001-7545-593X

## Ethics
All experiments were performed in accordance with the guidelines for maintenance and care of animals of the Canadian Council on Animal Care (CCAC) and approved by the Institutional Committee for the Protection of Animals (CIPA) at the Centre Hospitalier de de l'Université de Montréal. protocol number CM20031CMRs.

Reviewer #1 (Public review): https://doi.org/10.7554/eLife.99988.3.sa1
Reviewer #2 (Public review): https://doi.org/10.7554/eLife.99988.3.sa2
Reviewer #3 (Public review): https://doi.org/10.7554/eLife.99988.3.sa3
Author response https://doi.org/10.7554/eLife.99988.3.sa4

## Additional files

### Supplementary files
MDAR checklist

Supplementary file 1. Document containing detailed statistical tests used in each figure.

### Data availability
Source data is available on the Open Science Framework.

The following dataset was generated:

| Author(s) | Year | Dataset title | Dataset URL | Database and Identifier |
|---|---|---|---|---|
| Murphy-Royal C | 2025 | Guayasamin et al. eLife manuscript data | https://osf.io/zsw6j/ | Open Science Framework, 10.17605/OSF.IO/ZSW6J |

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
