## [Editor Report · eLife Assessment]

This **important** article explores the impact of early-life stress (ELS) on adult brain and behaviour. The significance of the **convincing** findings is that they implicate regulation of non-neuronal cells in the development of brain and behavioural dysfunction associated with ELS. With an elegant combination of behavioural models, morphological and functional assessments using immunostaining, electrophysiology, and viral-mediated loss-of-function approaches, the authors report that astrocyte dysfunction plays a role in ELS responses. The work is of interest to a broad behavioural and cellular neuroscience audience.

---

## [Referee Report · Reviewer #1 (Public review)]

Summary:

The manuscript asks the question of whether astrocytes contribute to behavioral deficits triggered by early life stress. This question is tested by experiments that monitor the effects of early life stress on anxiety-like behaviors, long-term potentiation in the lateral amygdala, and immunohistochemistry of astrocyte-specific (GFAP, Cx43, GLT-1) and general activity (c-Fos) markers. Secondarily, astrocyte activity in the lateral amygdala is impaired by viruses that suppress gap-junction coupling or reduce astrocyte Ca2+ followed by behavioral, synaptic plasticity, and c-Fos staining. Early life stress is found to reduce expression of GFAP, Cx43 and induce translocation of the glucocorticoid receptor to astrocytic nuclei. Both early life stress and astrocyte manipulations are found to result in generalization of fear to neutral auditory cues. All of the experiments are done well with appropriate statistics and control groups. The manuscript is very well-written and the data are presented clearly. The authors' conclusion that lateral amygdala astrocytes regulate amygdala-dependent behaviors is strongly supported by the data as is the conclusion that cellular and behavioral outcomes provoked by early life stress are similar to the outcomes provoked by astrocyte dysfunction. However, the extent to which early life stress requires astrocytes to generate these outcomes remains open to debate.

Strengths:

A strong combination of behavioral, electrophysiology, and immunostaining approaches is utilized and possible sex-differences in behavioral data are considered. The experiments clearly demonstrate that disruption of astrocyte networks or reduction of astrocyte Ca2+ provoke generalization of fear and impair long-term potentiation in lateral amygdala. The provocative finding that astrocyte dysfunction accounts for a subset of behavioral effects of early life stress (e.g. not elevated plus or distance traveled observations) is also perceived as a strength.

Weaknesses:

The main weakness is absence of direct evidence that behavioral and neuronal plasticity after early life stress can be attributed to astrocytes. It remains unknown what would happen if astrocyte activity were disrupted concurrently with early life stress or if changes in astrocyte Ca2+ could attenuate early life stress outcomes. As is, the only presented evidence that early life stress involves astrocytes is nuclear translocation of GR and downregulation of GFAP and Cx43 in Figure 3 which may or may not cause the reported astrocyte activity changes.

---

## [Referee Report · Reviewer #2 (Public review)]

Summary:

In this manuscript, Guayasamin et al. show that early-life stress (ELS) can induce a shift in fear generalisation in mice. They took advantage of a fear conditioning paradigm followed by a discrimination test and complement learning and memory findings with measurements for anxiety-like behaviors. Next, astrocytic dysfunction in the lateral amygdala was investigated at the cellular level by combining staining for c-Fos with astrocyte-related proteins. Changes in excitatory neurotransmission were observed in acute brains slices after ELS suggesting impaired communication between neurons and astrocytes. To confirm causality of astrocytic-neuronal dysfunction in behavioral changes, viral manipulations were performed in unstressed mice. Occlusion of functional coupling with a dominant negative construct for gap junction connexin 43 or reduction in astrocytic calcium with CalEx mimicked the behavioral changes observed after ELS suggesting that dysfunction of the astrocytic network underlies ELS-induced memory impairments.

Strengths:

Overall, this well written manuscript highlights a key role for astrocytes in regulating stress-induced behavioral and synaptic deficits in the lateral amygdala in the context of ELS. Results are innovative, and methodological approaches relevant to decipher the role of astrocytes in behaviors. As mentioned by the authors, non-neuronal cells are receiving increasing attention in the neuroscience, stress and psychiatry fields.

Weaknesses:

I did have several suggestions and comments that were addressed during the review process. I believe that it improved clarity and will increase the impact of the work.

---

## [Referee Report · Reviewer #3 (Public review)]

Summary:

The authors show that ELS induces a number of brain and behavioral changes in the adult lateral amygdala. These changes include enduring astrocytic dysfunction, and inducing astrocytic dysfunction via genetic interventions is sufficient to phenocopy the behavioral and neural phenotypes suggesting astrocyte dysfunction may play a causal role in ELS-associated pathologies.

Strengths:

A strength is the shift in focus to astrocytes to understand how ELS alters adult behavior.

Weaknesses:

The mechanistic links between some of the correlates - altered astrocytic function, changes in neural excitability and synaptic plasticity in the lateral amygdala and behavior - are underdeveloped.

Comments on revisions:

The authors have significantly improved the paper with the addition of new experimental data, analyses, and textual changes.

---

## [Author Response]

The following is the authors’ response to the original reviews.

**eLife Assessment**
Early-life adversity or stress can enhance stress susceptibility by causing changes in emotion, cognition, and reward-seeking behaviors. This important manuscript highlights the involvement of lateral amygdala astrocytes in fear generalization and the associated synaptic plasticity, which are parallel to the effects of early life stress. With an elegant combination of behavioral models, morphological and functional assessments using immunostaining, electrophysiology, and viral-mediated loss-of-function approaches, the authors provide solid correlational and causal evidence that is consistent with the hypothesis that early life stress produces neural and behavioral dysfunction via perturbing lateral amygdala astrocytic function.

We would like to thank the authors and editors for taking the time to review our work, and re-review it now. Also, we are grateful for this very positive assessment of our work. In this revised manuscript we made a strong effort to address comments made by all reviewers, providing clarification where required and new data to our manuscript in order to further support our observations.

**Public Reviews:**

**Reviewer #1 (Public Review):**
Summary:The manuscript asks the question of whether astrocytes contribute to behavioral deficits triggered by early life stress. This question is tested by experiments that monitor the effects of early life stress on anxiety-like behaviors, long-term potentiation in the lateral amygdala, and immunohistochemistry of astrocyte-specific (GFAP, Cx43, GLT-1) and general activity (c-Fos) markers. Secondarily, astrocyte activity in the lateral amygdala is impaired by viruses that suppress gap-junction coupling or reduce astrocyte Ca2+ followed by behavioral, synaptic plasticity, and c-Fos staining. Early life stress is found to reduce the expression of GFAP and Cx43 and to induce translocation of the glucocorticoid receptor to astrocytic nuclei. Both early life stress and astrocyte manipulations are found to result in the generalization of fear to neutral auditory cues. All of the experiments are done well with appropriate statistics and control groups. The manuscript is very well-written and the data are presented clearly. The authors' conclusion that lateral amygdala astrocytes regulate amygdala-dependent behaviors is strongly supported by the data. However, the extent to which astrocytes contribute to behavioral and neuronal consequences of early life stress remains open to debate.Strengths:A strong combination of behavioral, electrophysiology, and immunostaining approaches is utilized and possible sex differences in behavioral data are considered. The experiments clearly demonstrate that disruption of astrocyte networks or reduction of astrocyte Ca2+ provokes generalization of fear and impairs long-term potentiation in the lateral amygdala. The provocative finding that astrocyte dysfunction accounts for a subset of behavioral effects of early life stress (e.g. not elevated plus or distance traveled observations) is also perceived as a strength.Weaknesses:The main weakness is the absence of more direct evidence that behavioral and neuronal plasticity after early life stress can be attributed to astrocytes. It remains unknown what would happen if astrocyte activity were disrupted concurrently with early life stress or if the facilitation of astrocyte Ca2+ would attenuate early life stress outcomes. As is, the only evidence that early life stress involves astrocytes is nuclear translocation of GR and downregulation of GFAP and Cx43 in Figure 3 which may or may not provoke astrocyte Ca2+ or astrocyte network activity changes.

We would like to thank the reviewer for their constructive feedback on our work. In the revised version we have added new experiments that further support a role of astrocytes in ELS-induced behavioural dysfunction. Specifically, we carried out two-photon calcium imaging in lateral amygdala astrocytes using viral overexpression of membrane tethered GCaMP6f. These experiments revealed a decrease in astrocyte calcium activity following ELS (Figure 4). Interestingly these data also showed an important number of sex differences (Figure 4 - Figure supplement 1).

These new data allow us to strengthen the link between ELS-induced astrocyte hypofunction and behavioural changes. Indeed, we validated the impact of CalEx on astrocyte calcium activity in the lateral amygdala, again using two-photon microscopy, and show that CalEx resulted in an astrocyte calcium signature that very closely resembled that of ELS, i.e. reduced frequency and amplitude of events (Figure 5 - Figure supplement 2). As such, we feel like these data, while still correlative in nature, strengthen our findings and conclusion that astrocyte dysfunction alone is sufficient to recapitulate the effects of stress on excitability, synaptic function, and behaviour.

**Reviewer #2 (Public Review):**
Summary:In this manuscript, Guayasamin et al. show that early-life stress (ELS) can induce a shift in fear generalisation in mice. They took advantage of a fear conditioning paradigm followed by a discrimination test and complemented learning and memory findings with measurements for anxiety-like behaviors. Next, astrocytic dysfunction in the lateral amygdala was investigated at the cellular level by combining staining for c-Fos with astrocyte-related proteins. Changes in excitatory neurotransmission were observed in acute brains slices after ELS suggesting impaired communication between neurons and astrocytes. To confirm the causality of astrocytic-neuronal dysfunction in behavioral changes, viral manipulations were performed in unstressed mice. Occlusion of functional coupling with a dominant negative construct for gap junction connexin 43 or reduction in astrocytic calcium with CalEx mimicked the behavioral changes observed after ELS suggesting that dysfunction of the astrocytic network underlies ELS-induced memory impairments.Strengths:Overall, this well-written manuscript highlights a key role for astrocytes in regulating stress-induced behavioral and synaptic deficits in the lateral amygdala in the context of ELS. Results are innovative, and methodological approaches relevant to decipher the role of astrocytes in behaviors. As mentioned by the authors, non-neuronal cells are receiving increasing attention in the neuroscience, stress, and psychiatry fields.Weaknesses:I do have several suggestions and comments to address that I believe will improve the clarity and impact of the work. For example, there is currently a lack of information on the timeline for behavioral experiments, tissue collection, etc.

We thank the reviewer for their kind comments and constructive feedback on our manuscript. We agree that certain aspects could have been made more clear and we have revised the manuscript and figures to be more explicit regarding timelines. Including the addition of timelines on figures and improved clarity in the text where possible. We have also addressed the private comments provided by the reviewers alluded to in this public review.

**Reviewer #3 (Public Review):**
SummaryThe authors show that ELS induces a number of brain and behavioral changes in the adult lateral amygdala. These changes include enduring astrocytic dysfunction, and inducing astrocytic dysfunction via genetic interventions is sufficient to phenocopy the behavioral and neural phenotypes. This suggests that astrocyte dysfunction may play a causal role in ELS-associated pathologies.Strengths:A strength is the shift in focus to astrocytes to understand how ELS alters adult behavior.Weaknesses:The mechanistic links between some of the correlates - altered astrocytic function, changes in neural excitability, and synaptic plasticity in the lateral amygdala and behaviour - are underdeveloped.

We thank the reviewer for their comments. We are happy that they found our shift in focus towards astrocytes to be a strength of our work. Regarding mechanistic links being underdeveloped, we have attempted to address this by placing more effort into understanding the functional changes in astrocytes and how this relates to behaviour.

To address this comment we have used two-photon calcium imaging to quantify the impact of ELS on astrocyte calcium activity. As such, the revised manuscript contains several new figures including a detailed characterisation of the effects of ELS on astrocyte calcium activity (Figure 4), including sex differences in naive and the effects of stress (Figure 4 - Figure supplement 1), and an important validation of the impact of CalEx on astrocyte calcium activity. CalEx mirrors the impact of stress on astrocyte calcium activity reducing the frequency and amplitude of individual events (Figure 5 - Figure supplement 2).

Considering the strong overlap of the effects of ELS and CalEx on synapses, excitability, behaviour, and now astrocyte calcium activity, we hope that this added detail addresses some of the points highlighted by the reviewer.

**Recommendations for the authors:**
The reviewers all agree on one major issue for the authors to address. There is a bit of a lack of mechanistic linking between the astrocyte function and the early life stress and these data are more correlational than causal in nature. This could either be addressed by scaling back the data interpretation and title to be more reflective of the data at hand or if the authors would consider, doing the causal experiment of examining the manipulation of astrocyte activity following early life stress to see if this does influence the phenotype.

We agree with reviewers on this issue and realise that we have overstated our findings somewhat. As an immediate fix, suggested by reviewers, we have changed the title to more closely align with our data stating that astrocyte dysfunction is “associated with” rather than “induces” as well as adjusting our interpretations.

In addition to this one major comment, there are a list of minor comments that the authors should consider to improve the manuscript.(1) A major caveat is the lack of information on the timeline for behavioral experiments, tissue collection, etc. The authors mention "Mice between ages P45-70' but considering the developmental changes occurring between late adolescence and young adulthood, I recommend adding timelines on all Figures clearly indicating when behavioral tests were performed, and tissue collected for electrophysiology or immunostaining. With corticosterone (CORT) back at baseline at P70 vs a difference observed at P45 was this time point favored? It should be clarified throughout.

We apologise for the lack of clarity on this and have added more timelines on figures.

The age range favoured (p45-p70), relates to adolescence a time when latent psychiatric disorders tend to manifest in humans following early-life adversity. We have clarified this choice in the text.

(2) Given the transient increase in corticosterone levels in early-life stress mice, peaking at P45 and declining to control levels by P70, it would be informative to know whether the reported behavioral and synaptic changes differ within this time window. This may not be doable in the current approach, but this should be addressed nonetheless. Furthermore, it wasn't clear why the increase in blood corticosterone was delayed. Was this expected? How does this relate to earlier work? Wouldn't it be expected to be elevated at P17 (end of ELS period)?

We agree that this observation was very unexpected. Initially, we expected CORT to be elevated at P17, end of ELS period. We believe that low CORT levels during the ELS paradigm can be attributed to this paradigm coinciding with the stress hyporesponsive period (SHRP) which in rodents lasts until roughly postnatal day 14. During this period, mild stressors fail to elicit CORT responses. Considering our ELS paradigm lasts from P10-P17, there is a significant overlap with the SHRP.

This point is now included in the discussion with several citations regarding this biological phenomenon, as well as other studies that report similar findings to our own, i.e. a delayed increase in blood corticosterone levels following early-life stress.

(3) It is mentioned that behavioral tests were performed in both sexes with no sex differences observed. Were animals of both sexes also included in other experiments (ephys, immunostaining, blood CORT analysis)? Behavioral outcomes could be the same but underlying biological processes different. This is a topic that should be discussed. Identification of males vs females on graphs would be helpful.

We apologise for not having provided this data in the previous version of the manuscript. In the revised manuscript we provide analysis of sex differences for our initial behavioural observations (Figure 2 - Figure supplement 1), c-Fos (Figure 2 - Figure supplement 2), for GFAP and Cx43 (Figure 3 - Figure supplement 1), calcium signalling (Figure 4 - Figure supplement 1), and for CalEx and dnCx43 experiments across behaviour (Figure 5 - Figure supplement 4) and c-Fos (Figure 5 - Figure supplement 5).

(4) How long-lasting are the generalization phenotypes? Do they outlast the transient increase in blood corticosterone? Showing this would provide a more solid foundation for future explorations.

The reviewers raise a very important point. It remains unclear as to how long these effects last and this is something we are keen to address in future studies, with careful experiments designed to explicitly test this question, as well as subsequent questions regarding whether long-lasting effects are due to impaired brain development or whether these effects emerge due to CORT changes, or other changes, or a combination of them all?

As an aside, an additional manuscript from our lab (Depaauw-Holt et al. 2024 bioRxiv) which uses the same stressor but focuses on distinct brain regions and behaviours uses a prolonged time window in which the effects of stress are readily observable all the way to P90.

So while we do provide the answers in this work, it is a really great idea that we would like to follow up subsequently.

(5) With the ELS-induced change in locomotion, I would recommend presenting open field (center, periphery) and elevated plus maze (open, closed arms) data independently. It could also be interesting to analyze corner time in the open field as well as center time in the elevated plus maze.

We now provide data for the open field and elevated plus maze as requested. Our findings remain unchanged, but we agree with the reviewer that this way of representing the data is more clear.

(6) For Figure 2C, the ideal stats would be an ANOVA with CS (+/-) as a within-subject variable and treatment (naive/ELS) as a between-subjects variable. Then the best support for the generalization claim would be a CS x treatment interaction. I encourage the authors to do these stats. I note that this point is mitigated by the discrimination analysis presented in 2D (where they compare naive and ELS groups directly).

We have carried out the analysis as requested and these data further support the notion of fear generalisation in ELS mice (Figure 2 - Figure supplement 2A, B). Additionally, the analyses are included in a supplementary table. We hope that we have understood correctly, and this figure accurately reflects the reviewer’s suggestion.

(7) In Figure 2H, why not evaluate c-Fos levels after the discrimination test which is the main behavioral outcome? This statement in the Discussion should be modified if, as per my understanding, c-Fos was measured after the fear paradigm only "We find that both ELS and astrocyte dysfunction both enhance neuronal excitability, assessed by local c-Fos staining in the lateral amygdala following auditory discriminative fear conditioning. One interpretation of these data is that astrocytes might tune engram formation, with astrocyte dysfunction, genetically or after stress, increasing c-Fos expression resulting in a loss of specificity of the memory trace and generalisation of fear.'

We agree that further evaluation of c-Fos levels following the discrimination test would be insightful. We honestly did not consider this time point in our initial experimental design, as we considered previous reports in the literature that investigated how the numbers of cells recruited to the engram (c-Fos density) could influence memory accuracy at a later time point. As such, investigating c-Fos levels following training was our initial target. We have modified the text to be more explicit in our experimental approach.

This is nevertheless a fascinating point that we are keen to pursue in future studies.

(8) Some thoughts on why dnCx43 suppression of astrocyte network activity is less effective at inducing fear generalization than CalEx suppression of astrocyte Ca2+ are warranted. One might predict that both manipulations should result in similar effects, as seen in fEPSP and cFos data in Figure 4.

We agree that this is an interesting observation and the fact we did not observe the same behavioural phenotype despite fEPSP and c-Fos data to be the same is puzzling.

Nevertheless, we do see increased fear generalisation in both dnCx43 and CalEx. We hypothesise that CalEx had a more profound effect due to the wide range of processes that are presumably affected by reduced astrocyte calcium activity, whereas blocking gap junction channels still leaves a large number of astrocyte functions intact.

Overall, our conclusion is that behaviour is a more sensitive assay compared to the cellular phenotypes, which highlights the importance of answering these questions from multiple angles.

(9) Ideally changes in functional coupling following the dnCx43 manipulation should be shown here (line 169).

We, unfortunately, did not directly evaluate functional coupling in dnCx43 mice in this manuscript. This would have been a useful experiment, but we rely on our previous data where we extensively characterised this tool (Murphy-Royal et al. 2020 Nat Comms).

(10) It would be relevant to perform c-Fos staining with markers for astrocytes or neuronal cells. Is an increase in activity expected for both cell types?

This is a fascinating question, given recent work on this topic showing that astrocytes can indeed express c-Fos and may be recruited into engrams. We analysed our existing tissue, we found that indeed astrocytes were labelled with c-Fos following our behavioural conditioning paradigm. Our data align with recent reports, and we demonstrate a small percentage of astrocytes expressing c-Fos (Figure 2 - figure supplement 3). This modest number of astrocytes expressing c-Fos is discussed in the text and placed into context of very recent papers that have been published since our submission to eLife.

(11) Were the same mice subjected to behavior analysis than immunostaining?

We generated separate cohorts of mice for immunostaining and behaviour, and have made this more clear in the text.

(12) Language describing learning paradigm. The CS+ (line 73) isn't in itself aversive (and shouldn't be described as such). It acquires that value after pairing with the US (which is aversive).

We agree that this is poorly worded and have modified the text from “aversive cue” to “conditioned cue”.

(13) It is hard to appreciate the glucocorticoid receptor translocation with the images provided. Would it be possible to increase magnification or at least, provide small inserts at higher magnification?

We have re-imaged our brain sections to get more detailed images. These are provided in revised manuscript (Figure 3)

(14) For the viral injection experiment, for how long is the virus expressed before running behavior/recording/c-Fos staining? Is the age of the tested mice the same as Figures1-3 or they were injected at P45 and tested weeks later?

We age-matched all mice for all experiments and tried to keep our experimental window as tight as possible (p45-70). All mice were injected at P25-30 in order to meet the experimental time window. To be more precise we have added timelines on all figures.

(15) A validation of the virus is missing to confirm the reduction of Cx43 expression at mRNA and protein levels when compared to controls. A reference is provided but to my understanding age of the animals might be different.

Here, I believe the reviewer is referring to dnCx43. In this experiment we used a viral approach to overexpress a non-functional connexin 43 protein (Murphy-Royal et al. 2020 Nat Comms). As such, a PCR or immuno against this protein would be expected to reveal higher expression levels. We have tried to clarify this approach in the text.

It is true that we did not fully characterise this tool in the lateral amygdala which would have been useful but considering our extensive experience with this tool and in it’s development with our collaborators Baljit Khakh, Randy Stout, David Spray (see Murphy-Royal et al. 2020) we are confident in these data, despite the limitation of validation in this manuscript.

(16) Same comment for the CalEx, a validation would be appreciated. Based on Yu et al. could a GCaMP6f virus be more appropriate as control?

We agree this is an important experiment as our lab has not fully validated this tool in house (compared to dnCx43, which we previously validated).

As such, mice were injected with both a membrane tethered GCaMP6f under the control of the short GFAP promoter - AAV2/5-gfaABC1D-lck-GCaMP6f and CalEx - AAV2/5-gfaABC1D-hPMCA2w/b-mCherry. Using this approach we were able to record calcium activity from CalEx positive and CalEx negative astrocytes in the same tissue (Figure 5 - figure supplement 2).

We report that this approach does indeed reduce astrocyte calcium but does not entirely eliminate it. In fact, CalEx expressing astrocytes displayed similar calcium activity dynamics to that we observed following ELS. Together, this further strengthens our rationale to use CalEx in order to mimic the effects of stress on astrocytes, and determine downstream effects on excitability, synapses, and behaviour.

(17) Have previous studies found ELS generalization phenotypes in adulthood? If so, these should be discussed in more detail. If not, perhaps this point can be made more explicit.

This is a great point. After looking deeper into the literature in more depth we found an example of this in which ELS resulted in context fear generalisation in adult rats. This work is cited in the discussion in the context of our findings.

(18) A paper by Krugers et al (Biol Psychiatry 2020) seems especially relevant (glucocorticoids, fear generalization, engram size) and should be discussed.

Thank you for bringing this work to our attention. This is certainly important work that we had unfortunately overlooked. We have added a citation and discussed the manuscript Lesuis et al. Biol. Psychiatry 2021, which contains the data discussed in the conference proceeding by Krugers et al. Biol. Psychiatry 2020.

Additionally, we added another great manuscript by Lesuis et al. recently published in *Cell* in which they investigated the cellular mechanisms by which acute stress results in fear generalisation via endocannabinoids.

(19) Minor text revisions are necessary at lines 101 and 264 as well as p.5, line 58: "ratio" and p.10, line 128: "region of interest".

Thank you for pointing out these typos and errors. We have corrected them.

**Editor's note:**
Should you choose to revise your manuscript, please include full statistical reporting including exact p-values wherever possible alongside the summary statistics (test statistic and df) and 95% confidence intervals. These should be reported for all key questions and not only when the p-value is less than 0.05 in the main manuscript.